# Clebsch–Gordan Nets: a Fully Fourier Space Spherical Convolutional Neural Network

**Risi Kondor**[1*]   **Zhen Lin**[1*]   **Shubhendu Trivedi**[2*]
[1]The University of Chicago    [2]Toyota Technological Institute
{risi, zlin7}@uchicago.edu, shubhendu@ttic.edu

## Abstract

Recent work by Cohen *et al.* [1] has achieved state-of-the-art results for learning spherical images in a rotation invariant way by using ideas from group representation theory and noncommutative harmonic analysis. In this paper we propose a generalization of this work that generally exhibits improved performace, but from an implementation point of view is actually simpler. An unusual feature of the proposed architecture is that it uses the Clebsch–Gordan transform as its only source of nonlinearity, thus avoiding repeated forward and backward Fourier transforms. The underlying ideas of the paper generalize to constructing neural networks that are invariant to the action of other compact groups.

## 1 Introduction

Despite the many recent breakthroughs in deep learning, we still do not have a satisfactory understanding of how deep neural networks are able to achieve such spectacular perfomance on a wide range of learning problems. One thing that is clear, however, is that certain architectures pick up on natural invariances in data, and this is a key component to their success. The classic example is of course Convolutional Neural Networks (CNNs) for image classification [2]. Recall that, fundamentally, each layer of a CNN realizes two simple operations: a linear one consisting of convolving the previous layer's activations with a (typically small) learnable filter, and a nonlinear but pointwise one, such as a ReLU operator[2]. This architecture is sufficient to guarantee *translation equivariance*, meaning that if the input image is translated by some vector $t$, then the activation pattern in each higher layer of the network will translate by the same amount. Equivariance is crucial to image recognition for two closely related reasons: (a) It guarantees that exactly the same filters are applied to each part the input image regardless of position. (b) Assuming that finally, at the very top of the network, we add some layer that is translation *invariant*, the entire network will be invariant, ensuring that it can detect any given object equally well regardless of its location.

Recently, a number of papers have appeared that examine equivariance from the theoretical point of view, motivated by the understanding that the natural way to generalize convolutional networks to other types of data will likely lead through generalizing the notion of equivariance itself to other transformation groups [3, 4, 5, 6, 7]. Letting $f^s$ denote the activations of the neurons in layer $s$ of a hypothetical generalized convolution-like neural network, mathematically, equivariance to a group $G$ means that if the inputs to the network are transformed by some transformation $g \in G$, then $f^s$ transforms to $T_g^s(f^s)$ for some fixed set of linear transformations $\{T_g^s\}_{g \in G}$. s(Note that in some contexts this is called "covariance", the difference between the two words being only one of emphasis.)

---

[*]Authors are arranged alphabetically

[2]Real CNNs typically of course have multiple channels, and correspondingly multiple filters per layer, but this does not fundamentally change the network's invariance properties.

A recent major success of this approach are Spherical CNNs [1][8], which are an $\mathrm{SO}(3)$–equivariant neural network architecture for learning images painted on the sphere[3]. Learning images on the sphere in a rotation invariant way has applications in a wide range of domains from 360 degree video through drone navigation to molecular chemistry [9, 10, 11, 12, 13, 14, 15]. The key idea in Spherical CNNs is to generalize convolutions using the machinery of noncommutative harmonic analysis: employing a type of generalized $\mathrm{SO}(3)$ Fourier transform [16, 17], Spherical CNNs transform the image to a sequence of matrices, and compute the spherical analog of convolution in Fourier space. This beautiful construction guarantees equivariance, and the resulting network attains state of the art results on several benchmark datasets.

One potential drawback of Spherical CNNs of the form proposed in [1], however, is that the nonlinear transform in each layer still needs to be computed in "real space". Consequently, each layer of the network involves a forward and a backward $\mathrm{SO}(3)$ Fourier transform, which is relatively costly, and is a source of numerical errors, especially since the sphere and the rotation group do not admit any regular discretization similar to the square grid for Euclidean space.

Spherical CNNs are not the only context in which the idea of Fourier space neural networks has recently appeared [18, 19, 5, 7]. From a mathematical point of view, the relevance of Fourier theoretic ideas in all these cases is a direct consequence of equivariance, specifically, of the fact that the $\{T_g^s\}_{g \in G}$ operators form a *representation* of the underlying group, in the algebraic sense of the word [20]. In particular, it has been shown that whenever there is a compact group $G$ acting on the inputs of a neural network, there is a natural notion of Fourier transformation with respect to $G$, yielding a sequence of Fourier matrices $\{F_\ell^s\}_\ell$ at each layer, and the linear operation at layer $s$ will be equivariant to $G$ if and only if it is equivalent to multiplying each of these matrices from the right by some (learnable) filter matrix $H_\ell^s$ [7]. Any other sort of operation will break equivariance. The spherical convolutions employed in [1] are a special case of this general setup for $\mathrm{SO}(3)$, and the ordinary convolutions employed in classical CNNs are a special case for the integer translation group $\mathbb{Z}^2$. In all of these cases, however, the issue remains that the nonlinearities need to be computed in "real space", necessitating repeated forward and backward Fourier transforms.

In the present paper we propose a spherical CNN that differs from [1] in two fundamental ways:

1. While retaining the connection to noncommutative Fourier analysis, we relax the requirement that the activation of each layer of the network needs to be a (vector valued) function on $\mathrm{SO}(3)$, requiring only that it be expressible as a collection of some number of $\mathrm{SO}(3)$–covariant vectors (which we call fragments) corresponding to different irreducible representations of the group. In this sense, our architecture is strictly more general than [1].

2. Rather than a pointwise nonlinearity in real space, our network takes the tensor (Kronecker) product of the activations in each layer followed by decomposing the result into irreducible fragments using the so-called *Clebsch–Gordan decomposition*. This way, we get a "fully Fourier space" neural network that avoids repeated forward and backward Fourier transforms.

The resulting architecture is not only more flexible and easier to implement than [1], but our experiments show that it can also perform better on some standard datasets.

The Clebsch–Gordan transform has recently appeared in two separate preprints discussing neural networks for learning physical systems [21, 22]. However, to the best of our knowledge, it has never been proposed as a general purpose nonlinearity for covariant neural networks. In fact, any compact group has a Clebsch–Gordan decomposition (although, due to its connection to angular momentum in physics, the $\mathrm{SO}(3)$ case is by far the best known), so, in principle, the methods of the present paper could be applied much broadly, in any situation where one desires to build a neural network that is equivariant to some class of transformations captured by a compact group.

## 2 Convolutions on the sphere

The simplest example of a covariant neural network is a classical $S+1$ layer CNN for image recognition. In each layer of a CNN the neurons are arranged in a rectangular grid, so (assuming for simplicity that the network has just one channel) the activation of layer $s$ can be regarded as a function $f^s \colon \mathbb{Z}^2 \to \mathbb{R}$, with $f^0$ being the input image. The neurons compute $f^s$ by taking the

cross-correlation[4] of the previous layer's output with a small (learnable) filter $h^s$,

$$(h^s \star f^{s-1})(x) = \sum_y h^s(y - x) \, f^{s-1}(y), \tag{1}$$

and then applying a nonlinearity $\sigma$, such as the Re-LU operator:

$$f^s(x) = \sigma((h^s \star f^{s-1})(x)). \tag{2}$$

Defining $T_x(h^s)(y) = h^s(y - x)$, which is nothing but $h^s$ translated by $x$, allows us to equivalently write (1) as

$$(h^s \star f^{s-1})(x) = \langle f^{s-1}, T_x(h^s) \rangle, \tag{3}$$

where the inner product is $\langle f^{s-1}, T_x(h^s) \rangle = \sum_y f^{s-1}(y) \, T_x(h^s)(y)$. What this formula tells us is that fundamentally each layer of the CNN just does pattern matching: $f^s(x)$ is an indication of how well the part of $f^{s-1}$ around $x$ matches the filter $h^s$.

Equation 3 is the natural starting point for generalizing convolution to the unit sphere, $S^2$. An immediate complication that we face, however, is that unlike the plane, $S^2$ cannot be discretized by any regular (by which we mean rotation invariant) arrangement of points. A number of authors have addressed this problem in different ways [14, 15]. Instead of following one of these approaches, similarly to recent work on manifold CNNs [23, 24], in the following we simply treat each $f^s$ and the corresponding filter $h^s$ as continuous functions on the sphere, $f^s(\theta, \phi)$ and $h^s(\theta, \phi)$, where $\theta$ and $\phi$ are the polar and azimuthal angles. We allow both these functions to be complex valued, the reason for which will become clear later.

The inner product of two complex valued functions on the surface of the sphere is given by the formula

$$\langle g, h \rangle_{S_2} = \frac{1}{4\pi} \int_0^{2\pi} \int_{-\pi}^{\pi} [g(\theta, \phi)]^* \, h(\theta, \phi) \cos\theta \, d\theta \, d\phi, \tag{4}$$

where $*$ denotes complex conjugation. Further, $h$ (dropping the layer index for clarity) can be moved to any point $(\theta_0, \phi_0)$ on $S^2$ by taking $h'(\theta, \phi) = h(\theta - \theta_0, \phi - \phi_0)$. This suggests that the generalization of 3 to the sphere should be

$$(h \star f)(\theta_0, \phi_0) = \frac{1}{4\pi} \int_0^{2\pi} \int_{-\pi}^{\pi} [h(\theta - \theta_0, \phi - \phi_0)]^* \, f(\theta, \phi) \cos\theta \, d\theta \, d\phi. \tag{5}$$

Unfortunately, this generalization would be *wrong*, because it does not take into account that $h$ can also be rotated around a third axis. The correct way to generalize cross-correlations to the sphere is to define $h \star f$ as a function *on the rotation group itself*, i.e., to set

$$(h \star f)(R) = \frac{1}{4\pi} \int_0^{2\pi} \int_{-\pi}^{\pi} \left[ h_R(\theta, \phi) \right]^* f(\theta, \phi) \, \cos\theta \, d\theta \, d\phi \qquad R \in \mathrm{SO}(3), \tag{6}$$

where $h_R$ is $h$ rotated by $R$, expressible as

$$h_R(x) = h(R^{-1}x), \tag{7}$$

with $x$ being the point on the sphere at position $(\theta, \phi)$ (c.f. [25][1]).

## 2.1 Fourier space filters and activations

Cohen et al.[1] observe that the double integral in (6) would be extremely inconvenient to compute in a neural network. As mentioned, in the case of the sphere, just finding the right discretizations to represent $f$ and $h$ is already problematic. As an alternative, it is natural to represent both these functions in terms of their spherical harmonic expansions

$$f(\theta, \phi) = \sum_{\ell=0}^{\infty} \sum_{m=-\ell}^{\ell} \widehat{f}_\ell^m \, Y_\ell^m(\theta, \phi) \qquad h(\theta, \phi) = \sum_{\ell=0}^{\infty} \sum_{m=-\ell}^{\ell} \widehat{h}_\ell^m \, Y_\ell^m(\theta, \phi). \tag{8}$$

Here, $Y_\ell^m(\theta, \phi)$ are the well known spherical harmonic functions indexed by $\ell = 0, 1, 2, \ldots$ and $m \in \{-\ell, -\ell+1, \ldots, \ell\}$. The spherical harmonics form an orthonormal basis for $L_2(S^2)$, so (8) can be seen as a kind of Fourier series on the sphere, in particular, the elements of the $\widehat{f}_0, \widehat{f}_1, \widehat{f}_2, \ldots$ coefficient vectors can be computed relatively easily by

$$\widehat{f}_\ell^m = \frac{1}{4\pi} \int_0^{2\pi} \int_{-\pi}^{\pi} f(\theta, \phi) \, Y_\ell^m(\theta, \phi) \, \cos\theta \, d\theta \, d\phi,$$

and similarly for $h$. Similarly to usual Fourier series, in practical scenarios spherical harmonic expansions are computed up to some limiting "frequency" $L$, which depends on the desired resolution.

Noncommutative harmonic analysis [26, 27] tells us that functions on the rotation group also admit a type of generalized Fourier transform. Given a function $g\colon \mathrm{SO}(3) \to \mathbb{C}$, the Fourier transform of $g$ is defined as the collection of *matrices*

$$G_\ell = \frac{1}{4\pi} \int_{\mathrm{SO}(3)} g(R) \, \rho_\ell(R) \, d\mu(R) \qquad\qquad \ell = 0, 1, 2, \ldots, \tag{9}$$

where $\rho_\ell\colon \mathrm{SO}(3) \to \mathbb{C}^{(2\ell+1)\times(2\ell+1)}$ are fixed matrix valued functions called the irreducible representations of $\mathrm{SO}(3)$, sometimes also called Wigner D-matrices. Here $\mu$ is a fixed measure called the Haar measure that just hides factors similar to the $\cos\theta$ appearing in (4). For future reference we also note that one dimensional irreducible representation $\rho_0$ is the constant representation $\rho_0(R) = (1)$. The inverse Fourier transform is given by

$$g(R) = \sum_{\ell=0}^{\infty} \mathrm{tr}\left[ G_\ell \, \rho_\ell(R^{-1}) \right] \qquad\qquad R \in \mathrm{SO}(3).$$

While the spherical harmonics can be chosen to be real, the $\rho_\ell(R)$ representation matrices are inherently complex valued. This is the reason that we allow all other quantities, including the $f^s$ activations and $h^s$ filters to be complex, too.

Remarkably, the above notions of harmonic analysis on the sphere and the rotation group are closely related. In particular, it is possible to show that each Fourier component of the spherical cross correlation (6) that we are interested in computing is given simply by the outer product

$$[\widehat{h \star f}]_\ell = \widehat{f}_\ell \cdot \widehat{h}_\ell^\dagger \qquad\qquad \ell = 0, 1, 2, \ldots, L, \tag{10}$$

where $\dagger$ denotes the conjugate transpose (Hermitian conjugate) operation. Cohen et al.'s Spherical CNNs [1] are essentially based on this formula. In particular, they argue that instead of the continuous function $f$, it is more expedient to regard the components of the $\widehat{f}_0, \widehat{f}_1, \ldots, \widehat{f}_L$ vectors as the "activations" of their neural network, while the learnable weights or filters are the $\widehat{h}_0, \widehat{h}_1, \ldots, \widehat{h}_L$ vectors. Computing spherical convolutions in Fourier space then reduces to just computing a few outer products. Layers $s = 2, 3, \ldots, S$ of the Spherical CNN operate similarly, except that $f^{s-1}$ is a function on $\mathrm{SO}(3)$, so (6) must be replaced by cross-correlation on $\mathrm{SO}(3)$ itself, and $h$ must also be a function on $\mathrm{SO}(3)$ rather than just the sphere. Fortuitiously, the resulting cross-correlation formula is almost exactly the same:

$$[\widehat{h \star f}]_\ell = F_\ell \cdot H_\ell^\dagger \qquad\qquad \ell = 0, 1, 2, \ldots, L, \tag{11}$$

apart from the fact that now $F_\ell$ and $H_\ell$ are matrices (see [1] for details).

## 3   Generalized spherical CNNs

The starting point for our Generalized Spherical CNNs is the Fourier space correlation formula (10). In contrast to [1], however, rather than the geometry, we concentrate on its algebraic properties, in particular, its behavior under rotations. It is well known that if we rotate a spherical function by some $R \in \mathrm{SO}(3)$ as in (7), then each vector of its spherical harmonic expansion just gets multiplied with the corresponding Wigner D-matrix:

$$\widehat{f}_\ell \mapsto \rho_\ell(R) \cdot \widehat{f}_\ell. \tag{12}$$

For functions on $\mathrm{SO}(3)$, the situation is similar. If $g\colon \mathrm{SO}(3) \to \mathbb{C}$, and $g'$ is the rotated function $g'(R') = g(R^{-1}R')$, then the Fourier matrices of $g'$ are $G_\ell' = \rho_\ell(R) \, G_\ell$. The following proposition shows that the matrices output by the (10) and (11) cross-correlation formulae behave analogously.

**Proposition 1** *Let $f \colon S^2 \to \mathbb{C}$ be an activation function that under the action of a rotation $R$ transforms as (7), and let $h \colon S^2 \to \mathbb{C}$ be a filter. Then, each Fourier component of the cross correlation (6) transforms as*

$$[\widehat{h \star f}]_\ell \mapsto \rho_\ell(R) \cdot [\widehat{h \star f}]_\ell. \tag{13}$$

*Similarly, if $f', h' \colon \mathrm{SO}(3) \to \mathbb{C}$, then $\widehat{h' \star f'}$ (as defined in (11)) transforms the same way.*

Equation (12) describes the behavior of spherical harmonic *vectors* under rotations, while (15) describes the behavior of Fourier *matrices*. However, the latter is equivalent to saying that each column of the matrices separately transforms according to (12). One of the key ideas of the present paper is to take this property as the basis for the definition of covariance to rotations in neural nets. Thus we have the following definition.

---

**Definition 1** *Let $\mathcal{N}$ be an $S+1$ layer feed-forward neural network whose input is a spherical function $f^0 \colon S^2 \to \mathbb{C}^d$. We say that $\mathcal{N}$ is a **generalized** $\mathrm{SO}(3)$**–covariant spherical CNN** if the output of each layer $s$ can be expressed as a collection of vectors*

$$\widehat{f}^s = (\underbrace{\widehat{f}^s_{0,1}, \widehat{f}^s_{0,2}, \ldots, \widehat{f}^s_{0,\tau^s_0}}_{\ell=0}, \underbrace{\widehat{f}^s_{1,1}, \widehat{f}^s_{1,2}, \ldots, \widehat{f}^s_{1,\tau^s_1}}_{\ell=1}, \ldots\ldots\ldots, \underbrace{\ldots \widehat{f}^s_{L,\tau^s_L}}_{\ell=L}), \tag{14}$$

*where each $\widehat{f}^s_{\ell,j} \in \mathbb{C}^{2\ell+1}$ is a $\rho_\ell$–covariant vector in the sense that if the input image is rotated by some rotation $R$, then $\widehat{f}^s_{\ell,j}$ transforms as*

$$\widehat{f}^s_{\ell,j} \mapsto \rho(R) \cdot \widehat{f}^s_{\ell,j}. \tag{15}$$

*We call the individual $\widehat{f}^s_{\ell,j}$ vectors the irreducible **fragments** of $\widehat{f}^s$, and the integer vector $\tau^s = (\tau^s_0, \tau^s_1, \ldots, \tau^s_L)$ counting the number of fragments for each $\ell$ the **type** of $\widehat{f}^s$.*

---

There are a few things worth noting about Definition 1. First, since the (15) maps are linear, clearly any $\mathrm{SO}(3)$–covariant spherical CNN is equivariant to rotations, as defined in the introduction. Second, since in [1] the inputs are functions on the sphere, whereas in higher layers the activations are functions on $\mathrm{SO}(3)$, their architecture is a special case of Definition 1 with $\tau^0 = (1, 1, \ldots, 1)$ and $\tau^s = (1, 3, 5, \ldots, 2L+1)$ for $s \geq 1$.

Finally, by the theorem of complete reducibility of representations of compact groups, *any* $f^s$ that transforms under rotations linearly is reducible into a sequence of irreducible fragments as in (14). This means that (14) is really the most general possible form for an $\mathrm{SO}(3)$ equivariant neural network. As we remarked in the introduction, technically, the terms "equivariant" and "covariant" map to the same concept. The difference between them is one of emphasis. We use the term "equivariant" when we have the same group acting on two objects in a way that is qualitively similar, as in the case of the rotation group acting on functions on the sphere and on cross-correlation functions on $\mathrm{SO}(3)$. We use the term "covariant" if the actions are qualitively different, as in the case of rotations of functions on the sphere and the corresonding transformations (15) of the irreducible fragments in a neural network.

To fully define our neural network, we need to describe three things: 1. The form of the linear transformations in each layer involving learnable weights, 2. The form of the nonlinearity in each layer, 3. The way that the final output of the network can be reduced to a vector that is rotation *invariant*, since that is our ultimate goal. The following subsections describe each of these components in turn.

## 3.1 Covariant linear transformations

In a covariant neural network architecture, the linear operation of each layer must be covariant. As described in the Introduction, in classical CNNs, convolution automatically satisfies this criterion. In the more general setting of covariance to the action of compact groups, the problem was studied in [7]. The specialization of their result to our case is the following.

**Proposition 2** *Let $\widehat{f}^s$ be an $\mathrm{SO}(3)$–covariant activation function of the form (14), and $\widehat{g}^s = \mathcal{L}(\widehat{f}^s)$ be a linear function of $\widehat{f}^s$ written in a similar form. Then $\widehat{g}^s$ is $\mathrm{SO}(3)$–covariant if and only each $\widehat{g}^s_{\ell,j}$ fragment is a linear combination of fragments from $\widehat{f}^s$ with the same $\ell$.*

Proposition 2 can be made more clear by stacking all fragments of $\widehat{f}$ corresponding to $\ell$ into a $(2\ell+1) \times \tau_\ell^s$ dimensional matrix $F_\ell^s$, and doing the same for $\widehat{g}$. Then the proposition tells us simply that

$$G_\ell^s = F_\ell^s\, W_\ell^s \qquad\qquad \ell = 0, 1, 2, \ldots, L \qquad (16)$$

for some sequence of complex valued martrices $W_0^s, \ldots, W_L^s$. Note that $W_\ell^s$ does not necessarily need to be square, i.e., the number of fragments in $\widehat{f}$ and $\widehat{g}$ corresponding to $\ell$ might be different. In the context of a neural network, the entries of the $W_\ell^s$ matrices are learnable parameters.

Note that the Fourier space cross-correlation formulae (10) and (11) are special cases of (16) corresponding to taking $W_\ell = \widehat{h}_\ell^\dagger$ or $W_\ell = H_\ell^\dagger$. The case of general $W_\ell$ does not have such an intuitive interpretation in terms of cross-correlation. What the (16) lacks in interpretability it makes up for in terms of generality, since it provides an extremely simple and flexible way of inducing SO(3)–covariant linear transformations in neural networks.

## 3.2   Covariant nonlinearities: the Clebsch–Gordan transform

Differentiable nonlinearities are essential for the operation of multi-layer neural networks. Formulating covariant nonlinearities in Fourier space, however, is more challenging than formulating the linear operation. This is the reason that most existing group equivariant neural networks perform this operation in "real space". However, as discussed above, moving back and forth between real space and the Fourier domain comes at a signficiant cost and leads to a range of complications involving quadrature on the transformation group and numerical errors.

One of the key contributions of the present paper is to propose a fully Fourier space nonlinearity based on the Clebsch–Gordan transform. In representation theory, the Clebsch–Gordan decomposition arises in the context of decomposing the tensor (i.e., Kronecker) product of irreducible representations into a direct sum of irreducibles. In the specific case of SO(3), it takes form

$$\rho_{\ell_1}(R) \otimes \rho_{\ell_2}(R) = C_{\ell_1,\ell_2} \left[ \bigoplus_{\ell=|\ell_1-\ell_2|}^{\ell_1+\ell_2} \rho_\ell(R) \right] C_{\ell_1,\ell_2}^\top, \qquad\qquad R \in \mathrm{SO}(3),$$

where $C_{\ell_1,\ell_2}$ are fixed matrices. Equivalently, letting $C_{\ell_1,\ell_2,\ell}$ denote the appropriate block of columns of $C_{\ell_1,\ell_2}$,

$$\rho_\ell(R) = C_{\ell_1,\ell_2,\ell}^\top \left[ \rho_{\ell_1}(R) \otimes \rho_{\ell_2}(R) \right] C_{\ell_1,\ell_2,\ell}.$$

The CG-transform is well known in physics, because it is intimately related to the algebra of angular momentum in quantum mechanics, and the entries of the $C_{\ell_1,\ell_2,\ell}$ matrices can be computed relatively easily. The following Lemma explains why this construction is relevant to creating Fourier space nonlinearities.

**Lemma 3** *Let $\widehat{f}_{\ell_1}$ and $\widehat{f}_{\ell_2}$ be two $\rho_{\ell_1}$ resp. $\rho_{\ell_2}$ covariant vectors, and $\ell$ be any integer between $|\ell_1-\ell_2|$ and $\ell_1+\ell_2$. Then*

$$\widehat{g}_\ell = C_{\ell_1,\ell_2,\ell}^\top \left[ \widehat{f}_{\ell_1} \otimes \widehat{f}_{\ell_2} \right] \qquad (17)$$

*is a $\rho_\ell$–covariant vector.*

Exploiting Lemma 3, the nonlinearity used in our generalized Spherical CNNs consists of computing (17) between all pairs of fragments. In matrix notation,

$$G_\ell^s = \bigsqcup_{|\ell_1-\ell_2| \le \ell \le \ell_1+\ell_2} C_{\ell_1,\ell_2,\ell}^\top \left[ F_{\ell_1}^s \otimes F_{\ell_2}^s \right], \qquad (18)$$

where $\sqcup$ denotes merging matrices horizontally. Note that this operation increases the size of the activation substantially: the total number of fragments is squared, which can potentially be problematic, and is addressed in the following subsection.

The Clebsch–Gordan decomposition has recently appeared in two preprints discussing neural networks for learning physical systems [21, 22]. However, to the best of our knowledge, in the present context of a general purpose nonlinearity, it has never been proposed before. At first sight, the computational cost it would appear that the computational cost of (17) (assuming that $C_{\ell_1,\ell_2,\ell}$ has been precomputed) is $(2\ell_1+1)(2\ell_2+1)(2\ell+1)$. However, $C_{\ell_1,\ell_2,\ell}$ is actually sparse, in particular $[C_{\ell_1,\ell_2,\ell}]_{(m_1,m_2),m} = 0$ unless $m_1+m_2 = m$. Denoting the total number of scalar entries in the $F_{\ell\ \ell}^s$

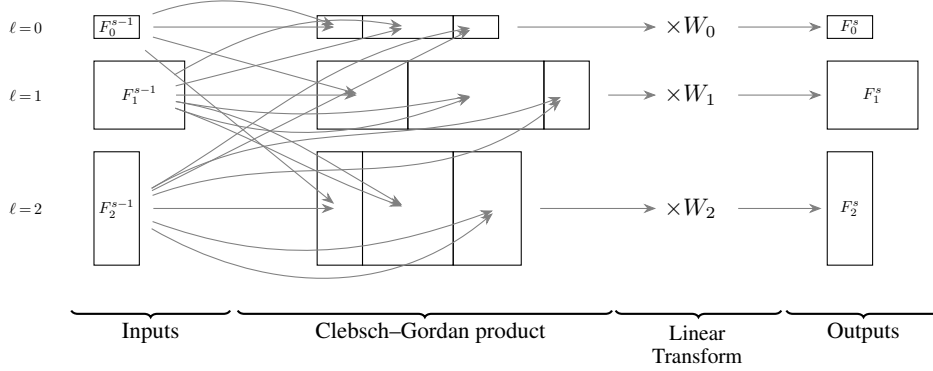

Figure 1: Schematic of a single layer of the Clebsch–Gordan network.

matrices by $N$, this reduces the complexity of computing (18) to $O(N^2 L)$. While the CG transform is not currently available as a differentiable operator in any of the major deep learning software frameworks, we have developed and will publicly release a C++ PyTorch extension for it.

A more unusual feature of the CG nonlinearity is that its essentially quadratic nature. Quadratic nonlinearities are not commonly used in deep neural networks. Nonetheless, our experiments indicate that the CG nonlinearity is effective in the context of learning spherical images. It is also possible to use higher CG powers, although the computational cost will obviously increase.

### 3.3 Limiting the number of channels

In a covariant network, each individual $\widehat{f}_\ell^s$ fragment is effecively a separate channel. In this sense, the quadratic increase in the number of channels after the CG-transform can be seen as a natural broadening of the network to capture more complicated features. Naturally, allowing the number of channels to increase quadratically in each layer would be untenable, though.

Following the results of Section 3.1, the natural way to counteract the exponential increase in the number of channels is follow the CG-transform with another learnable linear transformation that reduces the number of fragments for each $\ell$ to some fixed maximum number $\overline{\tau}_\ell$. In fact, this linear transformation can *replace* the transformation of Section 3.1. Whereas in conventional neural networks the linear transformation always precedes the nonlinear operation, in Clebsch–Gordan networks it is natural to design each layer so as to perform the CG-transform first, and then the convolution like step (16), which will limit the number of fragments.

### 3.4 Final invariant layer

After the $S-1$'th layer, the activations of our network will be a series of matrices $F_0^{S-1}, \ldots, F_L^{S-1}$, each transforming under rotations according to $F_\ell^{S-1} \mapsto \rho_\ell(R)\, F_\ell^{S-1}$. Ultimately, however, the objective of the network is to output a vector that is *invariant* with respect rotations, i.e., a collection of *scalars*. In our Fourier theoretic language, this simply corresponds to the $\widehat{f}_{0,j}^S$ fragments, since the $\ell=0$ representation is constant, and therefore the elements of $F_0^S$ are invariant. Thus, the final layer can be similar to the earlier ones, except that it only needs to output this single (single row) matrix.

Note that in contrast to other architectures such as [1] that involve repeated forward and backward transforms, thanks to their fully Fourier nature, for Clebsch–Gordan nets, in both training and testing, the elements of $F_0^S$ are guaranteed to be invariant to rotations of arbitrary magnitude not just approximately, but in the *exact* sense, up to limitations of finite precision arithmetic. This is a major advantage of Clebsch–Gordan networks compared to other covariant architectures.

### 3.5 Summary of algorithm

In summary, our Spherical Clebsch–Gordan network is an $S+1$ layer feed-forward neural network in which apart from the initial spherical harmonic transform, every other operation is a simple matrix operation. The algorithm is presented in explicit form in the Supplement.

| Method | RMSE |
|---|---|
| MLP/Random CM [28] | **5.96** |
| LGIKA (RF) [29] | 10.82 |
| RBF Kernels/Rand CM [28] | 11.42 |
| RBF Kernels/Sorted CM [28] | 12.59 |
| MLP/Sorted CM [28] | 16.06 |
| Spherical CNN [1] | 8.47 |
| Ours (FFS2CNN) | **7.97** |

| Method | P@N | R@N | F1@N | mAP | NDCG |
|---|---|---|---|---|---|
| Tatsuma_ReVGG | 0.705 | 0.769 | 0.719 | 0.696 | 0.783 |
| Furuya_DLAN | 0.814 | 0.683 | 0.706 | 0.656 | 0.754 |
| SHREC16-Bai_GIFT | 0.678 | 0.667 | 0.661 | 0.607 | 0.735 |
| Deng_CM-VGG5-6DB | 0.412 | 0.706 | 0.472 | 0.524 | 0.624 |
| Spherical CNNs [1] | 0.701 | 0.711 | 0.699 | 0.676 | 0.756 |
| FFS2CNNs (ours) | 0.707 | 0.722 | 0.701 | 0.683 | 0.756 |

Table 1: Results on the QM7 and 3D shape recognition datasets.

## 4 Experiments

In this section we describe experiments that give a direct comparison with those reported by Cohen *et al.* [1]. We choose these experiments as the Spherical CNN proposed in [1] is the only direct competition to our method.

**Rotated MNIST on the Sphere** We use a version of MNIST in which the images are painted onto a sphere and use two instances as in [1], more details about the data, baseline models, as well as the detailed architecture of our model and hyperparameters are provided in the appendix. We report three sets of experiments: For the first set both the training and test sets were not rotated (denoted NR/NR), for the second, the training set was not rotated while the test was randomly rotated (NR/R) and finally when both the training and test sets were rotated (denoted R/R).

| Method | NR/NR | NR/R | R/R |
|---|---|---|---|
| Baseline CNN | 97.67 | 22.18 | 12 |
| Cohen *et al.* | 95.59 | 94.62 | 93.4 |
| Ours (FFS2CNN) | 96.4 | 96 | 96.6 |

We observe that the baseline model's performance deteriorates in the three cases, more or less reducing to random chance in the R/R case. While our results are better than those reported in [1], they also have another characteristic: they remain roughly the same in the three regimes, while those of [1] slightly worsen. We think this might be a result of the loss of equivariance in their method.

**Atomization Energy Prediction** Next, we apply our framework to the QM7 dataset [30, 31], where the goal is to regress over atomization energies of molecules given atomic positions ($p_i$) and charges ($z_i$). Each molecule contains up to 23 atoms of 5 types (C, N, O, S, H). More details about the representations used, baseline models, as well as the architectural parameters are provided in the appendix. The final results are presented in the table, which show that our method outperforms the Spherical CNN of Cohen *et al.*. The only method that delivers better performance is a MLP trained on randomly permuted Coulomb matrices [28], and as [1] point out, this method is unlikely to scale to large molecules as it needs a large sample of random permutations, which grows rapidly with $N$.

**3D Shape Recognition** Finally, we report results for shape classification using the SHREC17 dataset [32], which is a subset of the larger ShapeNet dataset [33] having roughly 51300 3D models spread over 55 categories. Architectural details are provided in the appendix. We compare our results to some of the top performing models on SHREC (which use architectures specialized to the task) as well as the model of Cohen *et al.*. Our method, like the model of Cohen *et al.* is task agnostic and uses the same representation. Despite this, it is able to consistently come second or third in the competition, showing that it affords an efficient method to learn from spherical signals.

## 5 Conclusion

We have presented an $SO(3)$-equivariant neural network architecture for spherical data that operates completely in Fourier space, circumventing a major drawback of earlier models that need to switch back and forth between Fourier space and "real" space. We achieve this by – rather unconventionally – using the Clebsch-Gordan decomposition as the only source of nonlinearity. While the specific focus is on spheres and $SO(3)$-equivariance, the approach is more widely applicable, suggesting a general formalism for designing fully Fourier neural networks that are equivariant to the action of any compact continuous group.

## Footnotes

[3]$\mathrm{SO}(3)$ denotes the group of three dimensional rotations, i.e., the group of $3 \times 3$ orthogonal matrices.

[4] Convolution and cross-correlation are closely related mathematical concepts that are somewhat confounded in the deep learning literature. In this paper we are going to be a little more precise and focus on cross-correlation, because, despite their name, that is what CNNs actually compute.

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
