[Supplementary Material]

# Supplement to "Clebsch–Gordan Nets: a Fully Fourier Space Spherical Convolutional Neural Network"

**Risi Kondor**[1*] **Zhen Lin**[1*] **Shubhendu Trivedi**[2*]
[1]The University of Chicago    [2]Toyota Technological Institute
{risi, zlin7}@uchicago.edu, shubhendu@ttic.edu

## 1 Summary of algorithm

In summary our Spherical Clebsch–Gordan network is an $S+1$ layer feed-forward neural network with the following architecture. Note that assuming that the $C_{\ell_1,\ell_2,\ell}$ matrices have been precomputed, every operation in this algorithm apart from the initial spherical harmonic transform reduces to simple matrix operations.

1. The inputs to the network are $n_{\text{in}}$ functions $f_1^0, \ldots, f_{n_{\text{in}}}^0 \colon S^2 \to \mathbb{C}$. For example, for spherical color images, $f_1^0, f_2^0$ and $f_3^0$ might encode the red, green and blue channels. For generality, we treat these functions as complex valued, but of course they may also be real. The activation of layer $s = 0$ is the union of the spherical transforms of these functions $f_1^0, \ldots, f_{n_{\text{in}}}^0$ up to some band limit $L$, i.e.,

$$[\widehat{f}_{\ell,j}^0]_m = \frac{1}{4\pi} \int_0^{2\pi} \int_{-\pi}^{\pi} f_j^0(\theta, \phi)^* \, Y_\ell^m(\theta, \phi) \cos\theta \, d\theta d\phi. \tag{1}$$

   Therefore, the type of $\widehat{f}^0$ is $\tau^0 = (n_{\text{in}}, n_{\text{in}}, \ldots, n_{\text{in}})$, and $\widehat{f}^0$ is stored as a collection of $L+1$ matrices $\{F_0^0, F_1^0, \ldots, F_L^0\}$ of sizes $1 \times n_{\text{in}}, 3 \times n_{\text{in}}, 5 \times n_{\text{in}}, \ldots$

2. For layers $s = 1, \ldots, S-1$, the Fourier space activation $\widehat{f}^s = (F_0^s, F_1^s, \ldots, F_L^s)$ is computed as follows:

   (a) We form all possible Kronecker products

   $$G_{\ell_1,\ell_2}^s = F_{\ell_1}^{s-1} \otimes F_{\ell_2}^{s-1} \qquad 0 \le \ell_1 \le \ell_2 \le L.$$

   Note that the size of $G_{\ell_1,\ell_2}^s$ is $(2\ell_1+1)(2\ell_2+1) \times (\tau_{\ell_1}^{s-1}\tau_{\ell_2}^{s-1})$.

   (b) Each $G_{\ell_1,\ell_2}^s$ is decomposed into $\rho_\ell$–covariant blocks by $[G_{\ell_1,\ell_2}^s]_\ell = C_{\ell_1,\ell_2,\ell}^\top \, G_{\ell_1,\ell_2}^s$, where $C_{\ell_1,\ell_2,\ell}^\dagger$ is the rectangular Clebsch–Gordan matrix as in (17).

   (c) All $[G_{\ell_1,\ell_2}^s]_\ell$ blocks with the same $\ell$ are concatenated into a large matrix $H_\ell^s \in \mathbb{C}^{(2\ell+1) \times \overline{\tau_\ell^s}}$, and this is multiplied by the weight matrix $W_\ell^s \in \mathbb{C}^{\overline{\tau_\ell^s} \times \tau_\ell^s}$ to give

   $$F_\ell^s = H_\ell^s \, W_\ell^s \qquad\qquad \ell = 0, 1, \ldots, L.$$

3. The operation of layer $S$ is similar, except that the output type is $\tau^S = (n_{\text{out}}, 0, 0, \ldots, 0)$, so components with $\ell > 0$ do not need to be computed. By construction, the entries of $F_0^s \in \mathbb{C}^{1 \times n_{\text{out}}}$ are SO(3)–invariant scalars, i.e., they are invariant to simultaneous rotations of the input functions $f_1^0, \ldots, f_{n_{\text{in}}}^0 \colon S^2 \to \mathbb{C}$.

---

[*]Authors are arranged alphabetically

## 2 Experimental Details

Cohen *et al.* present two sets of experiments: In the first sequence, they study the numerical stability of their algorithm and quantify the equivariance error due to the quadrature. In the second, they present results on three datasets comparing with other methods. Since our method is fully equivariant, we focus on the second set of experiments.

### 2.1 Rotated MNIST on the Sphere

We use a version of MNIST in which the images are painted onto a sphere and use two instances as in [2]: One in which the digits are projected onto the northern hemisphere and another in which the digits are projected on the sphere and are also randomly rotated.

The baseline model is a classical CNN with $5 \times 5$ filters and 32, 64, 10 channels with a stride of 3 in each layer (roughly 68K parameters). This CNN is trained by mapping the digits from the sphere back onto the plane, resulting in nonlinear distortions. The second model we use to compare to is the Spherical CNN proposed in [2]. For this method, we use the same architecture as reported by the authors i.e. having layers $S^2$ convolution – ReLU – $SO(3)$ convolution – ReLU – Fully connected layer with bandwidths 30, 10 and 6, and the number of channels being 20, 40 and 10 (resulting in a total of 58K parameters).

For our method we use the following architecture: We set the bandlimit $L_{max} = 10$, and keep $\tau_l = \frac{12}{\sqrt{2l+1}}$, using a total of 5 layers as described in section 3.5, followed by a fully connected layer of size 256 by 10. We use a variant of batch normalization that preserves covariance in the fourier layers. This method takes a expanding average of the standard deviation for a particular fragment for all examples seen during training till then and divide the fragment by it (in testing, use the average from training); the parameter corresponding to the mean in usual batch normalization is kept to be zero as anything else will break covariance. Finally, we concatenate the output of each $F_0^s$ in each internal layer (length 24 each, as each is $\tau_0 = 12$ complex numbers), as well as the original coefficient at $l = 0$ (length 2), into a $SO(3)$ invariant vector of length 122. (We observed that having these skip connections was crucial to facilitate smooth training.) After that, we use the usual batch normalization [4] on the concatenated results before feeding it into fully connected layers of length 256, a dropout layer with dropout probability 0.5, and finally a linear layer to to 10 output nodes. The total number of parameters was 285772, the network was trained by using the ADAM optimization procedure [5] with a batch size of 100 and a learning rate of $5 \times 10^{-4}$. We also used L2 weight decay of $1 \times 10^{-5}$ on the trainable parameters.

### 2.2 Atomization Energy Prediction

We use the Coulomb Matrix (CM) representation proposed by [8], which is rotation and translation invariant but not permutation invariant. The Coulomb matrix $C \in \mathbb{R}^{N \times N}$ is defined such that for a pair of atoms $i \neq j$, $C_{ij} = (z_i z_j)/(|p_i - p_j|)$, which represents the Coulomb repulsion, and for atoms $i = j$, $C_{ii} = 0.5z_i^{2.4}$, which denotes the atomic energy due to charge. To test our algorithm we use the same set up as in [2]: We define a sphere $S_i$ around $p_i$ for each atom $i$. Ensuring uniform radius across atoms and molecules and ensuring no intersections amongst spheres during training, we define potential functions $U_z(x) = \sum_{j \neq i, z_j = z} \frac{ziz}{|x - p_i|}$ for every $z$ and for every $x$ on $S_i$. This yields a $T$ channel spherical signal for each atom in a molecule. This signal is then discretized using Driscol-Healy [3] grid using a bandwidth of $b = 10$. This gives a sparse tensor representation of dimension $N \times T \times 2b \times 2b$ for every molecule.

Our spherical CNN architecture has the same parameters and hyperparameters as in the previous subsection except that $\tau_l = 15$ for all layers, increasing the number of parameters to 1.1 M. Following [2], we share weights amongst atoms and each molecule is represented as a $N \times F$ tensor where $F$ represents $F_0^s$ scalars concatenated together. Finally, we use the approach proposed in [6] to ensure permutation invariance. The feature vector for each atom is projected onto 150 dimensions using a MLP. These embeddings are summed over atoms, and then the regression target is trained using another MLP having 50 hidden units. Both of these MLPs are jointly trained. The final results are presented below, which show that our method outperforms the Spherical CNN of Cohen *et al.*. The only method that delivers better performance is a MLP trained on randomly permuted Coulomb

matrices [7], and as [2] point out, this method is unlikely to scale to large molecules as it needs a large sample of random permutations, which grows rapidly with $N$.

## 2.3 3D Shape Recognition

We use the SHREC17 dataset [9], which is a subset of the larger ShapeNet dataset [1] having roughly 51300 3D models spread over 55 categories. It is divided into a 70/10/20 split for train/validation/test. Two versions of this dataset are available: A regular version in which the objects are consistently aligned and another where the 3D models are perturbed by random rotations. Following [2] we focus on the latter version, as well as represent each 3D mesh as a spherical signal by using a ray casting scheme. For each point on the sphere, a ray towards the origin is sent which collects the ray length, cosine and sine of the surface angle. In addition to this, ray casting for the convex hull of the mesh gives additional information, resulting in 6 channels. The spherical signal is discretized using the Discroll-Healy grid [3] with a bandwidth of 128. We use the code provided by [2] for generating this representation.

We use a ResNet style architecture, but with the difference that the full input is not fed back but rather different frequency parts of it. We consider $L_{max} = 14$, and first train a block only till $L = 8$ using $\tau_l = 10$ using 3 layers. The next block consists of concatenating the fragments obtained from the previous block and training for two layers till $L = 10$, repeating this process till $L_{max}$ is reached. These later blocks use $\tau_l = 8$. As earlier, we concatenate the $F_0^s$ scalars from each block to form the final output layer, which is connected to 55 nodes forming a fully connected layer. We use Batch Normalization in the final layer, and the normalization discussed in 2.1 in the Fourier layers. The model was trained with ADAM using a batch size of 100 and a learning rate of $5 \times 10^{-4}$, using L2 weight decay of 0.0005 for regularization.