[Reviews · NeurIPS 2018]

Reviewer 1



In this paper a deep convolutional neural network for learning spherical images is proposed by modifying the spherical CNNs of Cohen et al. using the Clebsch-Gordan transform. The main purpose is to improve the computing efficiency by using this transform instead of the Fourier transforms, while rotation invariance of spherical images is kept. Though some experiments are provided to demonstrate the proposed method, the computing efficiency is not fully illustrated.

Reviewer 2



The authors improve upon the Spherical CNNs work by Cohen et al [7] by proposing a neural network architecture that does not require forward-backward Fourier transforms at each layer while keeping the S(3) rotation invariance properties of theoutput. Instead all operations after the initial layer are in the frequency domain which allows the model to be more efficient according to the authors. I like the paper and introduction of new mathematical constructs into the neural networks domain. I also think that the paper is well written. I would have liked to see a complexity comparison w.r.t [7] in O() notation, in FLOPS and in actual CPU time for two architectures (one for [7] one for this paper) of similar accuracy. I think that the math is very involved and a simple table like that would emphasize the contribution in terms of speed up to a practitioner in a better way. Just a side comment, I think that the application for vision tasks on 3d models is interesting and the method itself is elegant. However, given that most of the sensor data is planar, how impactful do the authors think their architecture is? Also am i correct in saying that most sensor data is planar?

Reviewer 3



This paper proposes a generalized version of SO(3)-equivariant architectures including Spherical CNN. By utilizing the algebraic properties of Fourier transform and the tools in non-commutative harmonic analysis, the authors are able to construct (and prove) a most generalized version of SO(3)-equivariant architecture. Specifically, it only requires that, when an input image is rotated, each fragment (i.e., the output, Fourier coefficient vectors) of each layer will be multiplied by a Wigner-D matrix. To include non-linearities without performing inverse Fourier transform, the authors propose to use Clebsch-Gordon transformation. The experiments show that the proposed CG-Net can outperform Spherical CNN in several tasks. Strength: - The paper proposes a generalized version of Spherical CNN. Plus, the paper proves that this architecture is the most generalized version of SO(3)-equivariant architecture. - A principled treatment on SO(3)-equivarient operations by looking into the algebraic properties of them. As a result, almost all operations in CG-Net are performed in Fourier space. - CG-Net works well even without traditional non-linearity activation functions like ReLU. Weakness: - The improvement compared to Spherical CNN seems incremental. - The paper claims that it is costly to compute forward and backward Fourier transform in Spherical CNN. However, it does not compare the runtime of Spherical CNN and CG-Net. - The experiment seems unfair as the number of parameters in CG-Net is almost 5 times the number in Spherical CNN. - The paper does not include an ablation study on the hyperparameter tau, which is the maximal number of fragments for each spherical harmonic function index l in each layer (to avoid exponential growth in number of fragments) - It would be better if the authors include a visualization to provide a graphical explanation of the non-linearity in CG-transform. ---------------------- Authors' rebuttal only paritally addresses my concerns. I will keep the original rating